# Improved Point-Line Feature Based Visual SLAM Method for Complex Environments

**DOI:** 10.3390/s21134604

**Published:** 2021-07-05

**Authors:** Fei Zhou, Limin Zhang, Chaolong Deng, Xinyue Fan

**Affiliations:** 1College of Communication and Information Engineering, Chongqing University of Posts and Telecommunications, Chongqing 400065, China; zhoufei@cqupt.edu.cn (F.Z.); s190131177@stu.cqupt.edu.cn (C.D.); fanxy@cqupt.edu.cn (X.F.); 2Intelligent Terminal Key Laboratory of Sichuan Province, Yibin 644000, China

**Keywords:** visual SLAM, point and line feature, adaptive ORB, data association, LSD feature extraction, reprojection error

## Abstract

Traditional visual simultaneous localization and mapping (SLAM) systems rely on point features to estimate camera trajectories. However, feature-based systems are usually not robust in complex environments such as weak textures or obvious brightness changes. To solve this problem, we used more environmental structure information by introducing line segments features and designed a monocular visual SLAM system. This system combines points and line segments to effectively make up for the shortcomings of traditional positioning based only on point features. First, ORB algorithm based on local adaptive threshold was proposed. Subsequently, we not only optimized the extracted line features, but also added a screening step before the traditional descriptor matching to combine the point features matching results with the line features matching. Finally, the weighting idea was introduced. When constructing the optimized cost function, we allocated weights reasonably according to the richness and dispersion of features. Our evaluation on publicly available datasets demonstrated that the improved point-line feature method is competitive with the state-of-the-art methods. In addition, the trajectory graph significantly reduced drift and loss, which proves that our system increases the robustness of SLAM.

## 1. Introduction

In recent years, the application of mobile robots in the fields of service robots, industrial automatic guided vehicles (AGVs) [1], and unmanned driving have put forward higher requirements for autonomous navigation. Among them, SLAM [2] is the key technology to realize positioning and map construction, such as laser SLAM [3], visual SLAM (VSLAM) [4], and so on. Because of the advantages of low price and convenient portability, the visual SLAM method has become the mainstream technology for robot navigation today. Although the theoretical framework of the visual SLAM algorithm is relatively complete, the robustness of the algorithm is inadequate to meet the needs of the actual environment, and system crashes caused by data association failures occur frequently. Therefore, improving the robustness of data association in the SLAM algorithm is important.

Since Klein et al. introduced the parallel tracking and mapping (PTAM) [5] algorithms in 2007, many real-time visual SLAM methods have been proposed, which can be divided into two categories: direct methods and indirect methods. The direct methods directly use the pixel intensity to match the pixels in the image and perform pose estimation by minimizing the reprojection error, such as large-scale direct monocular SLAM (LSD-SLAM) [6], direct sparse odometry (DSO) [7], and direct sparse mapping (DSM) [8]; these methods are extremely sensitive to light. On the contrary, the indirect methods estimate the pose by tracking the point features of the image, such as oriented FAST and rotated BRIEF-SLAM (ORB-SLAM) [9] and RGBD-SLAM-V2 [10]. The indirect methods based on point features have long been considered the mainstream method for the front-end of SLAM [11] because they have the advantages of stability and insensitivity to light and dynamic objects. In addition, the indirect methods are also relatively mature solutions at present.

Point features are the lowest level feature, their related theories are relatively well developed, and they are easy to extract in rich-texture scenes. However, in complex environments such as low-texture environment, point features are less likely to maintain mutuality, and they may even disappear temporarily. These phenomena may lead to tracking failure, which often happen in man-made scenes. However, despite the lack of reliable feature points, by using more structural information from the real environment, such as line features, the system can obtain more accurate camera pose estimation. In addition, lines have conditional advantages; they are not so sensitive to scenes such as low-texture environment, wide viewing angle range, and motion blur, which are the main disadvantages of SLAM systems that only use point features. As shown in Figure 1, it can be found that there are far more line features in the scene than point features, and line features can better represent image information. Therefore, algorithms based on line features have received more and more attention [12,13,14,15].

However, the correspondence relationship based only on line features is not as reliable as the corresponding relationship based on point features because of the weakness of lines, and the cumulative global error generated by the long-term operation of the system will affect the performance of the system. In addition, the calculation and matching of line features are time-consuming, which will affect the real-time performance of the system. Therefore, the point and line features fusion has been applied to the SLAM system. Albert Pumarola et al. added line features for visual tracking based on ORB-SLAM, and they proposed a visual point and line SLAM (PL-SLAM) [16] algorithm combined with monocular points and lines. The line segments are represented by their endpoints, respectively, and the projection errors are calculated from the endpoints to the lines, which still use point features in essence. Xie Xiaojia et al. [17] improved the representation method of spatial lines. For the first time, they used orthogonal representation as a minimization parameter to simulate line features and point features in visual SLAM, analyzed the Jacobian matrix of the reprojection errors of line parameters, improved the SLAM solutions, and solved the problem of over-parameterization of lines. However, when constructing the errors optimization cost function [18], they simply added the reprojection errors of the points and lines, and did not consider the problems of line features that are easily prone to matching errors and other issues. In addition, these algorithms are only improved line features, and do not consider that point features are susceptible to sudden light changes.

In response to the above problems, this paper adds line features on the basis of ORB-SLAM and constructs an improved point-line integrated monocular VSLAM algorithm. The main contributions of this article are as follows:An adaptive ORB uniform distribution algorithm is proposed. We adjust the threshold adaptively according to the brightness of each image block. The extracted features are more uniform, and the algorithm is more robust to light mutation.The traditional LSD algorithm is improved. Our algorithm completes line features matching based on the results of point features matching. In addition, we also use merging line segments and eliminating short lines to shorten time and improve the correct rate of line features matching.A new cost function of reprojection error is proposed. We weight the reprojection errors of point and line features based on the richness and dispersion of features in the scene, so as to give full play to the advantages of point and line features.

The rest of this paper is structured as follows: Section 2 reviews the related work. Section 3 presents a brief description of our system. Section 4 introduces the improved point-line feature processing. Section 5 presents experimental results using datasets, followed by the conclusion in Section 6.

## 2. Related Work

The existing SLAM methods based on the point features are generally divided into two parts: front-end estimation and back-end optimization. The front-end extracts features from environmental information and matches them to complete the robot’s motion state estimation. The back-end uses filter methods or optimization methods to optimize the motion state calculated by the front-end. Point features methods have certain robustness to light and dynamic objects. For example, ORB-SLAM can achieve more robust camera tracking and relocation in scenes with rich textures. However, in the face of sudden light changes and low textures, the algorithm cannot associate or even detect point features, which directly leads to the failure of the algorithm.

There are many existing point features extraction algorithms, such as scale invariant feature transform (SIFT) [19], speeded up robust features (SURF) [20], oriented FAST and rotated BRIEF (ORB) [21]. SIFT and SURF have more accurate feature extraction results, but they are too time-consuming [22,23,24]. The open-source system ORB-SLAM uses ORB algorithm to extract features because ORB addresses the problem that the FAST corner points have no directionality, and it uses the extremely fast binary descriptor binary robust independent elementary feature (BRIEF) [25], which greatly accelerates the entire image features extraction process. ORB features are composed of two parts: FAST keypoints and BRIEF descriptors. FAST algorithm mainly detects the obvious changes in the gray level of local pixels. The idea of the algorithm is: if the gray value of a pixel and its neighboring pixels are too different (too bright or too dark), then it is more likely to be a corner point. Before the comparison, it is necessary to set the brightness comparison threshold between pixels in advance. The setting of the threshold is very important. On one hand, if the value is too small, the extracted corner points are not special enough and may also be “clustered”. On the other hand, if the value is too large, a sufficient number of point features may not be extracted. In order to solve the problem of uneven feature distribution caused by improper threshold selection, many scholars have done research. In ORB-SLAM system, Mur-Arta et al. [9] proposed the use of quadtree to improve the uniformity of feature distribution, but this was only to optimize the extraction results; this method essentially used the original FAST algorithm. Fan Xinnan et al. [26] proposed an adaptive threshold extraction method, but it still contained artificially set parameters and could not achieve true adaptive extraction.

Except for point features, line features and plane features appear frequently in the environment, where line features are more discriminative than point features and can directly provide geometric information of the scene. SLAM based on line features has attracted people’s attention for a long time. In 1997, the paper [27] proposed to apply vertical line segments to the SLAM system based on the extended Kalman filter (EKF) framework, but the line features detection results were not stable. Line features do not have complete extraction, description, and matching methods like point features. For a long time, the extraction and description of line features themselves have directly restricted the application of line features in SLAM [28]. Later, with the proposal of line segments detection methods such as line segment detector (LSD) [29] and line segments description methods such as line band descriptor (LBD) [30], line features are more widely used in SLAM systems.

As shown in Figure 2, compared with point features, the line features are more robust to the low-texture environment and the suddenly changing lightness environment. However, line features are less robust to scale and perspective. These lead to phenomena such as breakage of detected line features and added difficulty to the subsequent line features matching. Based on the problem of increased error and system breakdown caused by poor line features matching accuracy in the monocular SLAM system, this paper uses point features matching results to improve the accuracy of lines matching and introduces the weighted fusion method of point and line features to effectively avoid the impact of low matching accuracy of line features and improve the robustness and operating efficiency of the system after fusing point and line features in monocular SLAM.

## 3. System Overview

In this section, a brief description of our system is presented. Our constructed system adds line features information based on ORB-SLAM structure, which can make the line features work in feature tracking and local map construction. The overall flow of the algorithm is shown in Figure 3.

The system is completed by three parallel threads which can be seen from Figure 3 and the main improvements have been marked in the flowchart. After the initialization is completed, the system will maintain a local map composed of point and line features. In the tracking thread, whenever an image is input, the system uses dual threads to extract point and line features separately, calculates feature depth information through triangulation, and maps it to the local map. Then, with the help of the matching relationship between features, the system calculates the pose of the current frame. By reprojecting the features of the local map to the current frame, the reprojection error is calculated; at the same time, the system will determine whether the current frame is judged as a keyframe [31] according to the keyframe screening strategy.

The local map is composed of keyframes, map points, and map lines corresponding to the keyframes. This thread is responsible for maintaining local map information, including the insertion and deletion of keyframes, the insertion and deletion of corresponding features, and the local pose optimization using beam adjustment based on reprojection errors. Then, according to the similarity between the new keyframe and the previous keyframe, the system decides whether to delete some keyframes with a high coincidence rate.

After initialization is completed, the system will perform global loop closure detection [32] continuously. Loop closure detection has high requirements for feature density, illumination invariance, and scale invariance. In these aspects, point features have more advantages than line features. In addition, the computational cost of line features is too high, which will affect the real-time performance of the system. Therefore, this paper does not add line features to the loop closure detection part. For each feature, the system calculates the bag-of-words vector [33] of the feature. For each keyframe, the system calculates its visual vector and stores them in the keyframe database. When performing loop closure detection, the system searches for a keyframe in the keyframe database. If the keyframe is similar to the current frame in terms of visual vector, it will be regarded as a candidate frame for loop closure. Then, the system conducts further screening by introducing similar transformations to confirm whether there is a loop closure. If it exists, the system will bind the map points corresponding to the current frame and the candidate keyframe for global optimization.

In recent point-line combined SLAM system, the combination of LSD line segments extraction and LBD line segments description is usually used to complete the work of line features matching. Besides, the point-line features extraction and matching work are completely separated, which consumes a lot of computing resources. Moreover, these SLAM systems simply use the existing point and line features extraction algorithm, and have not improved the shortcomings of the algorithm. In view of the above problems, this paper improves the front end of ORB-SLAM system, and the main process is shown in Figure 4. The main improvements made in this paper have been marked in purple.

When a new image frame is input, the system performs point and line features extraction. First, the system uses ORB algorithm with adaptive threshold to extract point features in the image, and calculates the rough pose matrix of the current frame based on the point features matching results obtained. Then, based on this, the system performs preliminary screening of line segments to be matched, and then completes the line features matching. After obtaining the matching results of point and line features, the system reprojects the points and lines in the local map to the current frame, calculates the errors between the observed points and lines and the points and lines obtained by the reprojection, and sets reasonable values for the errors. Finally, the system optimizes the pose by minimizing the reprojection error.

## 4. Improved Point-Line Feature Processing

In this section, we first introduce the construction process of the ORB extractor with local adaptive threshold. Then we describe the process of line features extraction and matching, including line features extraction, line features preprocessing, and line features matching assisted by point features matching results. Finally, in Section 4.3, we show the weight construction model of the reprojection error of the point-line features.

### 4.1. Adaptive Point Feature Extractor

ORB detector can quickly extract pixels with obvious brightness difference in the image, so it is widely used in the field of visual SLAM. However, the point features extracted by the traditional ORB algorithm are unevenly distributed, and they are prone to “clustering”. Point features are redundant in regions with rich textures but rare in regions with weak textures. This will have a negative impact on visual SLAM, and can easily lead to failure of subsequent image matching, which will affect the accuracy of pose estimation. In response to this problem, Mur-Arta et al. [9] proposed a quadtree-based point features extraction algorithm, which divides image into uniform grids. Then, their system performed point features detections and descriptor calculations in the grids. Finally, the extracted image point features are evenly distributed. Figure 5 visually shows the point features extraction process. The main improvements made in this paper have been marked in blue in Figure 5.

The FAST algorithm performs features extraction by verifying whether the grayscale differences between the pixel to be detected and its surrounding pixels meet a certain threshold. In FAST, the threshold is manually determined, which usually depends on engineering experience. However, the point features extraction method based on a fixed threshold cannot be applied to all images, especially images with large differences in brightness. This will cause uneven distribution of point features, and the number of extracted point features will change sharply with the change of brightness. For this reason, inspired by the concept of “coefficient of variation” in mathematics, this paper proposes an adaptive threshold method based on gray information for automatic selection of FAST features extraction threshold. The algorithm automatically designs and calculates the threshold according to the brightness of the pixel. The calculation method is as follows:(1)$$iniTh_{j}=\sqrt{\frac{1}{n}\sum_{i=1}^{n}{\left(I_{j}\left(x_{i}\right)-\bar{I_{j}\left(x\right)}\right)}^{2}}/\bar{I_{j}\left(x\right)}$$
(2)$$minTh_{j}=iniTh_{j}/3$$
where $I_{j}\left(x_{i}\right)$ is the gray value of the *i*-th pixel in the *j*-th image block, $\bar{I_{j}\left(x\right)}$ is the average gray value of the *j*-th image block, and $iniTh_{j}$ is the calculated initial extraction threshold of the image block. In addition, referring to “dual threshold” idea coming from Mur-Arta, our system uses $minTh_{j}$ to extract again for areas where no point features can be extracted using the initial threshold. If this area is still not able to extract point features, then this area will be discarded.

In order to accurately estimate the pose, the selected features should be evenly distributed throughout the image. It is not appropriate to use the brightness of the entire image to determine the threshold for extracting features, because different areas of the image have different brightness, so selecting features in this way may cause the features selected in the low-textured grid to be unsuitable for tracking. For this reason, inspired by the empirical formula summarized by Tiller et al. [34], this paper uses the “average area occupied by each point” to divide the image block. The calculation formula for the side length of the image block is as follows:(3)$$w=\sqrt{width\times height/N_{\alpha }}$$
where height and width represent the height and width of the input image, respectively, $N_{\alpha }$ is the number of features required for the α-th pyramid image, s is the scale factor of the pyramid, n is the total number of pyramid layers, and the calculation formula for $N_{\alpha }$ is as follows:(4)$$N_{\alpha }=\frac{N\left(1-s\right)}{1-s^{n}}s^{\alpha }$$

### 4.2. Line Features Extraction and Matching

In this section, we first introduce the process of line features extraction and preprocessing, including the merging of broken lines and near lines, and the elimination of short lines. After that, we show the line features matching process assisted by the point features matching results in Section 4.2.2.

#### 4.2.1. Line Features Extraction and Preprocessing

In this paper, we used the famous LSD algorithm to extract the line features in the image and the LSD algorithm can extract the set of pixels with similar gradient directions in the image. This process can be completed in linear time without parameter adjustment. So, it is suitable for the line features detection work in this paper.

However, the LSD algorithm usually splits a long line segment into several short line segments. In addition, some detected line segments may be very near. The existence of such detected line segments usually complicates the subsequent lines matching task, which increases the uncertainty of line segments triangulation [35]. In order to minimize the impact of multiple line segments produced by one line, this paper uses three parameters: direction difference, point-line distance, and endpoint distance to merge near lines and eliminate short lines. Among them, the point-line distance is defined as the minimum distance l between the endpoints of two line segments and the vertical distance d from the midpoint of a line segment to another line segment. Figure 6 visually shows the calculation process of the point-line distance.

After finishing the above processing, the system then screens the remaining line segments according to their length, and ignores the line segments whose length is less than the threshold D. Figure 7 shows the line segments merging process and the effect of the improved line features extraction algorithm on the EuRoC dataset sequence MH_01_easy.

#### 4.2.2. Point Features Matching Results Assisted Line Features Matching

Many papers use LBD descriptors and introduce the geometric properties of lines to match the calculated line features. So, the matching methods used in many papers are very similar. For example, paper [17] mentioned that if two lines $l_{1}$ and $l_{2}$ can be successfully matched, then they need to meet the following conditions: (1) The angle difference between two lines is less than the threshold Φ. (2) The lengths of two lines are similar: $\min\left(\left\|l_{1}\right\|,\left\|l_{2}\right\|\right)/\max\left(\left\|l_{1}\right\|,\left\|l_{2}\right\|\right)>\tau$. (3) The overlap of two lines is greater than the threshold $l_{overlap}/\min\left(\left\|l_{1}\right\|,\left\|l_{2}\right\|\right)>\beta$. (4) The descriptor distance of two lines is less than the threshold ε.

The screening process of the above four conditions is extremely time-consuming, and the SLAM system is prone to losing real-time performance. In addition, the second filter condition is not reasonable. Because line segments often change in length due to changes in perspective or scale, this is the reason why many papers do not use the endpoints of line segments for matching. Therefore, in our system, we did not use the second filter condition.

Considering that the processing speed of the point features is faster than the line features, and the reliability of the point feature matching results is greater than that of the line features, this paper combines the matching result of the point features with the line segments to be matched when matching the line segments, that is, adding the angle limitation factor. The specific methods are as follows:

(1) For any two line segments, the system filters the line segments to be matched by the rotation angle between frames. A rough rotation angle can be obtained through the results of point features matching between adjacent frames. Suppose the rotation angle is θ, and then our system calculates the rotation angle between line segments to be matched and denoted it as $\theta_{l}$. Suppose the rotation angle error threshold is set to $\theta_{t}$ and set as $\theta_{t}$ = 6° during the experiment. If $\left|\theta -\theta_{l}\right|\leq \theta_{t}$ is satisfied, the next check operation can be performed.

(2) We traverse the candidate line segment and record the optimal and suboptimal matching distance of the descriptor, which are recorded as $d_{1}$ and $d_{2}$ respectively. If $d_{1}\leq 50$ and $d_{1}/d_{2}\geq 0.8$, the line segment is matched successfully. It is considered that two line segments do not match, and there is no need to perform descriptor matching.

This not only improves the accuracy of line features matching, but also speeds up the matching process.

### 4.3. Point-Line Features Error Weighted Model

When a new image is input, the system first extracts point and line features. This paper uses ORB point features extraction algorithm and LSD line features extraction algorithm, and then uses matching methods such as descriptors, optimal suboptimal ratios, and angle thresholds. These methods of features extraction and matching have been relatively complete, and the processing process can be referred to [36], which will not be described here. After obtaining the matching results of point and line features, the points and lines in the local map are reprojected to the current frame, and the pose is optimized by minimizing the reprojection error.

The reprojection error formulas of point and line features are defined as follows:(5)$$e_{p}=x-x^{\prime }$$
(6)$$e_{l}=d\left(z,l_{c}\right)={\left[\frac{p_{s}^{T}l_{c}}{\sqrt{l_{1}^{2}+l_{2}^{2}}},\frac{p_{e}^{T}l_{c}}{\sqrt{l_{1}^{2}+l_{2}^{2}}}\right]}^{T}$$
where x and $x^{\prime }$ represent the projection point and observation point of the three-dimensional space point, respectively, and z and $l_{c}$ represent the projection line and the observation line of the three-dimensional space line segment, respectively.

When constructing the error function, this paper introduces two criteria: the degree of the richness and dispersion of features, which are used as a benchmark to weight the results from two kinds of features. The former weights the score proportionally according to the number of features of a certain type (point or line) in a set of features detected in the image, while the latter takes the dispersion of features in the image into account (the greater the dispersion, the higher the weight); the error weights of points and lines are obtained by the following formulas:(7)$$w_{p}=0.5\left(\frac{n_{k}}{n_{k}+n_{l}}+\frac{d_{k}}{d_{k}+d_{l}}\right)$$
(8)$$w_{l}=0.5\left(\frac{n_{l}}{n_{k}+n_{l}}+\frac{d_{l}}{d_{k}+d_{l}}\right)$$
where $n_{k}$ and $n_{l}$ are the number of key points and lines extracted from the image, respectively, and the degree of dispersion of key points and lines ($d_{k}$ and $d_{l}$ respectively) are calculated by the square root of the sum of the variances of the x and y coordinates of the extracted features. In the case of lines, such x and y coordinates are taken from their midpoints. Finally, the optimized function is:(9)$$T=\arg\min\left[\sum_{i\in P}e_{p_{i}}^{T}\sum_{e_{p_{i}}}^{-1}w_{p}e_{p_{i}}+\sum_{j\in L}e_{l_{j}}^{T}\sum_{e_{l_{j}}}^{-1}w_{l}e_{l_{j}}\right]$$
where $e_{p_{i}}$ represents the reprojection error of the i-th pair of matching points, $e_{l_{j}}$ represents the reprojection error of the j-th matching line, and P and L represent the sets of matching points and matching lines, respectively.

## 5. Results

In this section, we first show the effectiveness of the improved point features extraction algorithm under uniformity, extraction speed, and different lighting intensity conditions in Section 5.1. Then, we show the positive effect of adding the improved point and line features to the system.

The experiment first uses the EuRoC [37] dataset to conduct experiments on the improved ORB algorithm, and then evaluates the system designed in this paper on the TUM [38] dataset and the KITTI [39] road dataset; at the same time, the experiment in Section 5.3 compares this system with the most advanced methods, including ORB-SLAM, PL-SLAM, LSD-SLAM, and RGBD-SLAM. All experiments are run in a 64-bit Linux operating system. The CPU of the running platform is I9-7900X, and the platform has 16G running memory.

### 5.1. ORB Homogenization Algorithm Based on Adaptive Threshold

In order to quantify the distribution of features, this paper uses the distribution uniformity function [40] to calculate the distribution uniformity of the point features. As shown in Figure 8, we divide the image from vertical, horizontal, 45°, 145°, center and periphery, so we obtain 10 regions of top, bottom, left, right, top left, bottom right, top right, bottom left, center, and periphery. Then we count the number of feature points in each area, calculate the variance V of this set of data, and the final uniformity u calculation formula is:(10)u=101×log(V)

We use Equation (10) to calculate the uniformity of each frame. The smaller the uniformity value, the better the uniform distribution effect.

In order to verify the effectiveness of our improved ORB algorithm in improving the distribution uniformity and computational efficiency, we used images from the KITTI dataset sequence 00. This sequence contains images with different viewing angles and different brightness. This experiment uses the feature extraction algorithm proposed by Mur-Arta in ORB-SLAM (in the following, it will be referred to as the MA algorithm for short) and the improved algorithm proposed in this paper to conduct comparative experiments, and for the sake of generality, all experiments are performed five times for each frame, and the average value of the five times of experiment results is taken as the final experiment result. The experiment results are shown in Table 1.

It can be seen from Table 1 that the algorithm proposed in this paper is better than the MA algorithm in the uniformity of the distribution, and the uniformity of the improved algorithm is increased by 13.06% on average. The time consumption is also significantly reduced, and time consumption of our improved algorithm is reduced by 54.42% on average. Next, we conduct experiments on pictures from real indoor scenes, desktop and lockers, to visualize the extraction effects of two algorithms. Figure 9 represents the extraction results of two kinds of extraction algorithms. It is obvious that the features extracted by our improved algorithm in this paper are more evenly distributed across the entire image.

In order to further verify the adaptability of our improved ORB algorithm under different brightness, we conducted experiments on the number of point features and feature repetition rate extracted by the improved algorithm under different lighting conditions. The specific operations are as follows: Firstly, we preprocess the original image to obtain new image sequences. Based on the experimental original image, the range of brightness change is −60% to 60% with 20% as an interval. Then we perform features extraction and results calculation on each changed image. Figure 10 shows the number of extracted point features and feature repetition rate under different brightness.

As shown in Figure 10a, it is obvious that the number of point features extracted by the improved ORB algorithm is generally lower than that of the MA algorithm, because the MA algorithm has a large number of overlapping point features. The number of point features extracted by the MA algorithm decreases sharply with the change of brightness, while the number of point features extracted by the improved algorithm does not change significantly, indicating that the improved algorithm has stronger adaptability to changes in brightness. Furthermore, the number of point features cannot accurately test the effectiveness of our algorithm, because compared with the error extraction and elimination caused by a fixed threshold, the adaptive threshold does not see a significant change in the number of point features. In order to further illustrate the advantages of our improved algorithm, repeatability is selected as a quantitative evaluation index. Repeatability is that the system can extract the same point features as the original image on the new image obtained after changing the brightness:(11)$$r=N_{r}/N_{f}$$
where $N_{r}$ is the number of point features appearing in the same position of two images; $N_{f}$ is the number of point features detected in the original image.

Figure 10b is the result of the repetition rate under the change of image brightness. It can be seen that the overall repetition rate of the improved ORB point features extraction algorithm is the highest, and the change with brightness is small, while the repetition rate of the MA algorithm decreases sharply after the brightness increases and decreases. This is enough to observe the effectiveness of the improved algorithm in this paper.

### 5.2. EuRoC Dataset Evaluation

This section uses the MH_01_easy scene sequence images in the EuRoC dataset. In this sequence, each scene contains images taken by the left and right sides of the binocular camera. Because this paper only makes improvements to the monocular ORB-SLAM, we all choose the images taken by the camera on the left. Since this dataset provides real trajectory information, we use root mean squared error (RMSE) to evaluate the accuracy of the running trajectory.

The calculation method of the RMSE is shown in the following formula:(12)$$RMSE\left(T\right)=\sqrt{\frac{\sum_{i=1}^{n}{\left\|T_{e,i}-T_{s,i}\right\|}^{2}}{n}}i$$
where $T_{e,i}$ and $T_{s,i}$ represent the estimated pose and real pose of the moving robot at time i respectively.

We compare our algorithm with ORB-SLAM and several state-of-the-art methods, including DSO [7], LDSO [41], PL-SLAM [42], and DSM [8]. The results of these advanced methods are from related papers, and ORB-SLAM is obtained by running open source code. When conducting experiments, each group of experiments is run more than five times. Finally, the median of the results of multiple runs is taken as the final experimental result. All experimental results are shown in Table 2.

Obviously, the accuracy of our system is higher than ORB-SLAM and PL-SLAM, which can be expected due to our better integration of point and line features. Among the six sequences listed, our system performed well in the medium and difficult sequences because of the weak-texture regions in these sequences. The point features of the easy sequences are relatively rich, which leads to the fact that the line features we added do not make outstanding contributions to our system. At the same time, DSM adds a map reuse module based on point features; this may be the reason why our system did not show the best results in the easy sequences. In addition, in the environment where the illumination changes significantly, such as MH_03_medium, the accuracy of DSO and LDSO based on the principle of constant luminosity decreases significantly. Overall, our system performs well on the EuRoC dataset.

In order to further illustrate the advantages of line features, Figure 11 shows the comparison of the posture error trajectory between the ORB-SLAM and the improved PL-SLAM in this paper under sequence V2_03_difficult. The sequence contains a weak-texture environment for a period of time. It can be seen that ORB-SLAM system obviously fails to locate, and it has not been able to relocate successfully during this period of time; what is shown on the picture is a straight track that deviates from the true value. However, the improved PL-SLAM in this paper can get close to the true value during this period, and the error is also small.

### 5.3. TUM Dataset Evaluation

During the image acquisition process of the TUM mono dataset, mobile robots will walk around in different rooms and corridors, and collect scenes such as walls, roofs, walkways, desktops, and table bottoms. This dataset contains scenes with rich features and weak textures. The types of scenes in the image sequence are relatively rich, the changes between consecutive image frames are relatively large, and the lighting changes are obvious. In this section, we selected several sequences with typical conditions such as image blur and repeated texture for testing, and compared the results with excellent systems such as LSD-SLAM.

The test standard used in the experiment is the RGB-D test standard provided by the Technical University of Munich and the experimental results are shown in Table 3. The experimental results of LSD-SLAM and other algorithms used for comparison in this section are from [16]. The experiment uses the RMSE to investigate the operating effects of each system. The results show that compared with other algorithms, the proposed algorithm has better accuracy and robustness in environments with weak textures and obvious illumination changes. Because the image sequences used contain obvious lighting changes and weak-texture environments, our method performs better than ORB-SLAM and RGBD-SLAM based only on point features, and better than LSD-SLAM based only on line features. Compared with PL-SLAM, our method reasonably integrates point and line features instead of simply adding errors, which can effectively improve the accuracy of feature matching and the speed of feature tracking in the point-line SLAM system. This may be the reason why our system performs better than PL-SLAM.

### 5.4. KITTI Dataset Evaluation

This section shows the results of ORB-SLAM and our system on sequence 00 and 06 of the KITTI dataset. The KITTI dataset is collected by autonomous driving vehicles, and the images obtained are outdoor public roads. As shown in Figure 12, orange and blue are the trajectories of our system and ORB-SLAM on different sequences, respectively. On the trajectory map in the bottom left corner of the 00 sequence, ORB-SLAM deviates from the true trajectory for a period of time. On the contrary, our system is very successful both on sequence 00 and 06, and the actual running trajectories almost completely coincide with the real trajectories.

Furthermore, in order to better verify the effectiveness and robustness of the improved system in this paper, we compare ORB-SLAM and our improved PL-SLAM with real trajectories on the x, y, z axis and yaw, pitch, and roll, respectively. We still use 00 and 06 sequences. As shown in Figure 13, orange and blue are the trajectories of our system and ORB-SLAM on different sequences, respectively. It is easy to see that the ORB-SLAM system has large errors in certain places on the 00 sequence, which may be due to the presence of multiple weak-texture regions in this sequence. In addition, the 06 sequence includes rich textures, so the trajectories of two systems are close to the real trajectories. In general, our system has smaller trajectory errors on each axis than ORB-SLAM and is closer to the true trajectory. This shows that the improvements made in this paper better solve the positioning accuracy problem of the original system.

## 6. Conclusions

In this paper, we introduce line features to ORB-SLAM system. Compared with the existing visual SLAM systems based on the combination of point and line features, our system not only considers how to better introduce line features, but also optimizes point features to improve its extraction efficiency and stability of responding to obvious brightness changes. In addition, considering the difference in the number and density of point and line features, this paper introduces reprojected error weighted model when constructing the optimization cost function. The more the number of features and the more scattered the distribution, the greater the reliability. Experiments show that compared with the state-of-the-art methods, our method has achieved good results in terms of accuracy. In addition, trajectory graph also proved that fusing these two types of features will produce more robust estimation in different datasets. In the future, we will investigate how to introduce inertial sensors into our system with point and line features.

## Figures and Tables

**Figure 1 sensors-21-04604-f001:**
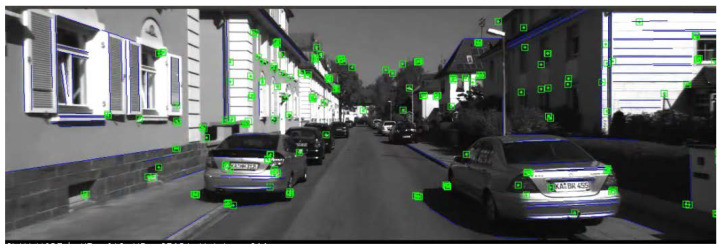
The results of point and line features on the KITTI dataset sequence 00. The green boxes represent the point features extracted from the image, and the blue lines are the extracted line features.

**Figure 2 sensors-21-04604-f002:**
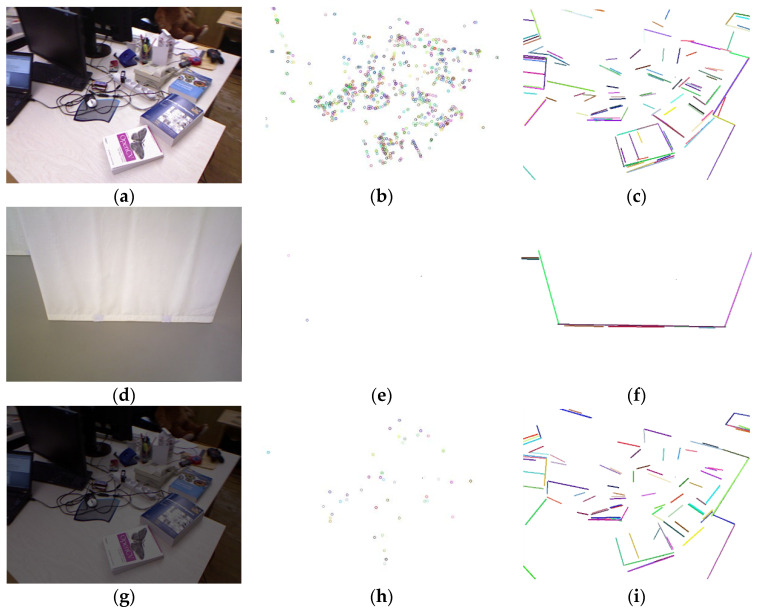
The performance of point and line features in areas of rich texture, low texture, and sudden changes in brightness. (**a**,**d**,**g**) A rich-texture scene, low-texture scene, and scene with brightness reduced to 40% of (**a**). We can clearly see that the ORB point features (**b**,**e**,**h**) and LSD line features (**c**,**f**,**i**) are extracted in the above scene.

**Figure 3 sensors-21-04604-f003:**
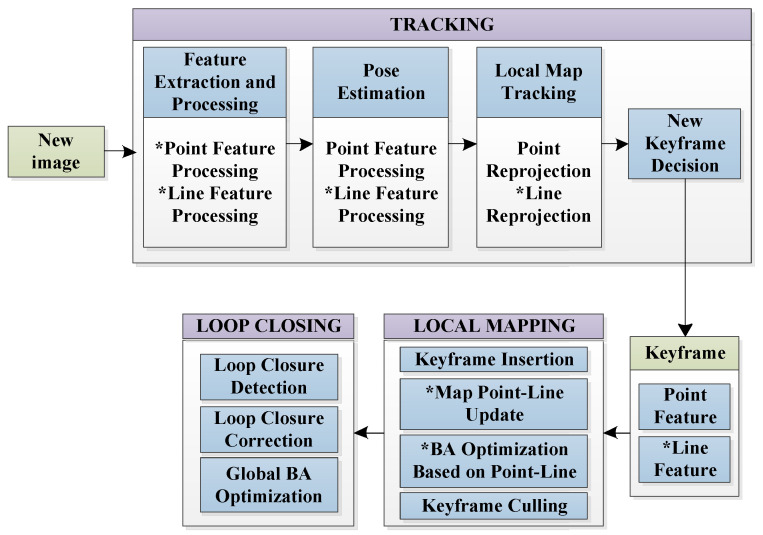
System overview. Our system pipeline is an extension of ORB-SLAM. The system is completed by three parallel threads: tracking, local mapping, and loop closure. Compared with ORB-SLAM, the main improvements of our system have been marked with “*” in the flowchart.

**Figure 4 sensors-21-04604-f004:**
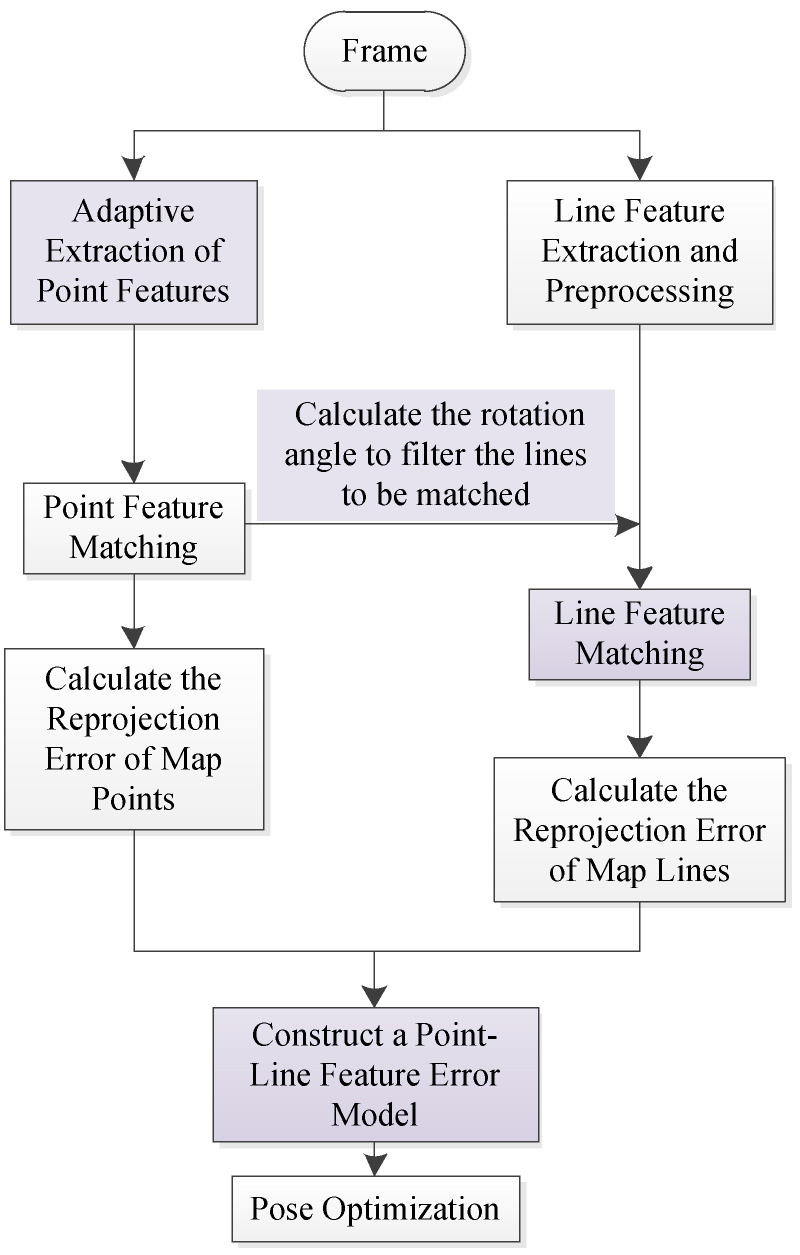
The front-end framework diagram based on the improved point-line features algorithm.

**Figure 5 sensors-21-04604-f005:**
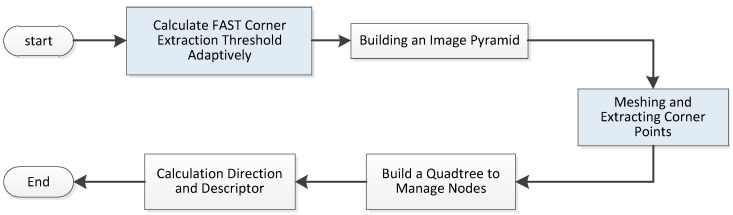
Flow chart of improved ORB algorithm.

**Figure 6 sensors-21-04604-f006:**
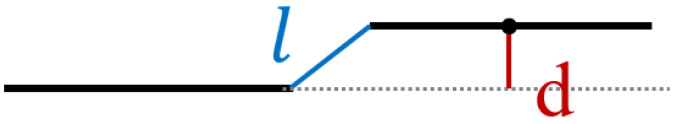
Point-line distance between two line segments.

**Figure 7 sensors-21-04604-f007:**
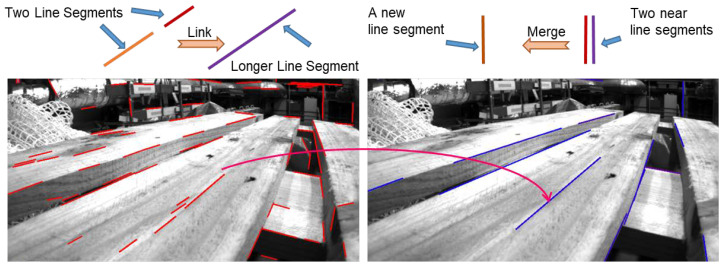
We link line segments into a longer one, and merge near line segments into a new one to enhance the quality of lines extracted by LSD. The image comes from EuRoC dataset sequence MH_01_easy.

**Figure 8 sensors-21-04604-f008:**
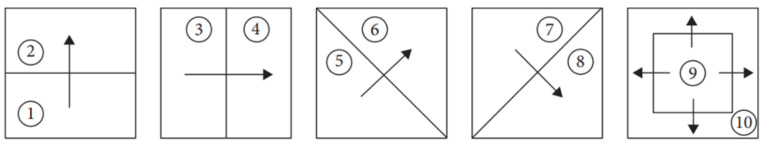
Details of the 10 areas divided when calculating the distribution uniformity.

**Figure 9 sensors-21-04604-f009:**
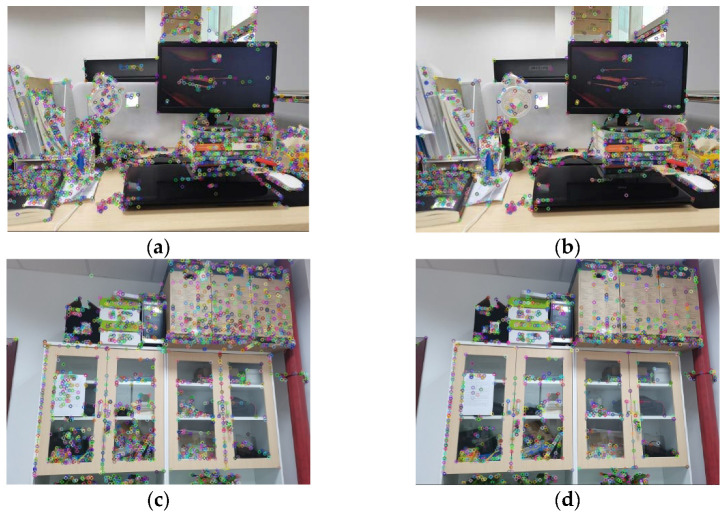
The extraction results of two algorithms on real indoor scenes: (**a**,**b**) desktop images, (**c**,**d**) locker images. Images (**a**,**c**) show the distribution of the point features extracted by the MA algorithm, and images (**b**,**d**) show the distribution of the feature points extracted by the improved algorithm in this paper. The colored circles are the extracted point features.

**Figure 10 sensors-21-04604-f010:**
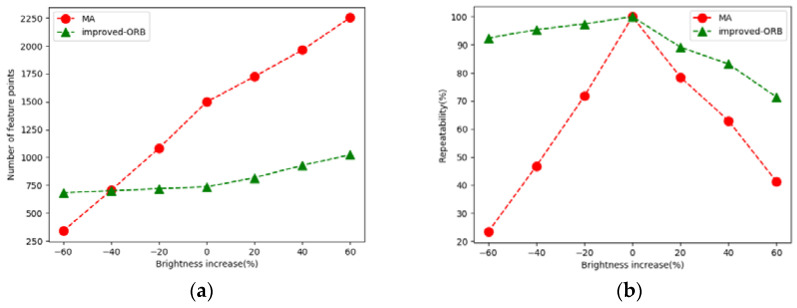
The performance of two algorithms in the case of changes in brightness. (**a**,**b**) They show the comparison of the number of point features and the feature repetition rates under different lighting conditions, respectively. The green lines show the advantages of the improved algorithm in this paper.

**Figure 11 sensors-21-04604-f011:**
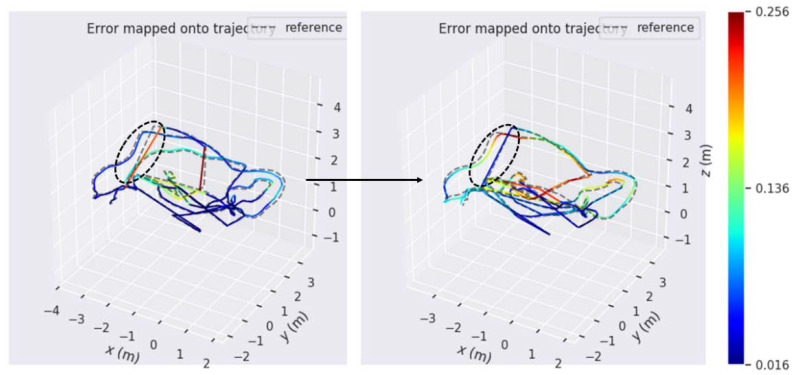
Comparison of the pose error trajectory of two algorithms under the sequence V2_03_difficult. The color of the trajectory changes from blue to red, and the closer the color is to red, the bigger the error. The one on the left is the posture trajectory error map based on ORB-SLAM, and the one on the right is the posture trajectory error map based on the improved PL-SLAM. Circled in black are trajectories of two algorithms corresponding to a continuous weak-texture environment in the sequence.

**Figure 12 sensors-21-04604-f012:**
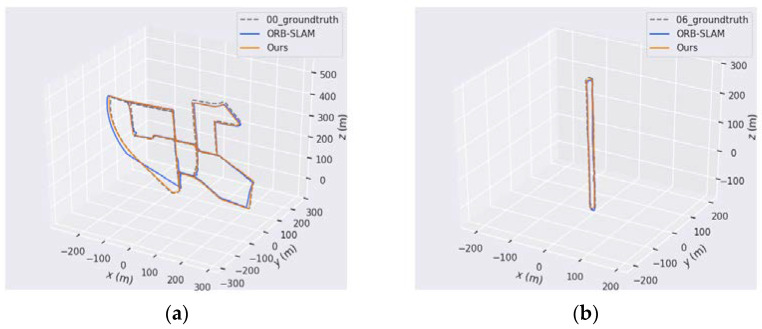
Performance of different algorithms on the different sequences of the KITTI dataset. (**a**) is the 00 sequence with multiple loops, (**b**) is the 06 sequence with long road and one loop.

**Figure 13 sensors-21-04604-f013:**
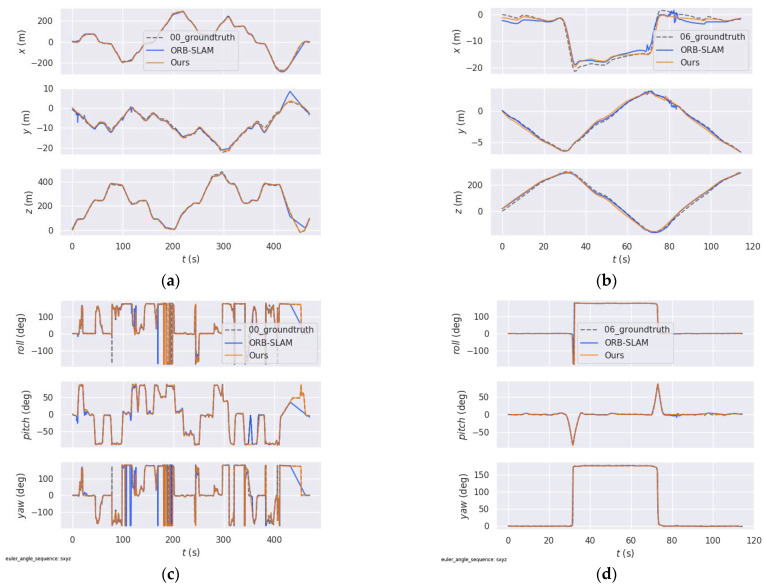
Performance of different algorithms on the different sequences of the KITTI dataset. (**a**,**b**) are the trajectory errors of different algorithms on the X, Y, and Z axis. (**c**,**d**) are the trajectory errors of different algorithms on the yaw, pitch, and roll.

**Table 1 sensors-21-04604-t001:** Comparison of the uniformity and time consumption of two algorithms on the KITTI dataset sequence 00.

Image Sequence	Distribution Uniformity		Time Consumption/ms	
	MA Algorithm	Improved Algorithm	MA Algorithm	Improved Algorithm
1	389.84	**325.91**	2.543	**1.318**
2	518.85	**465.88**	5.255	**2.705**
3	495.95	**404.86**	3.701	**1.143**
4	487.64	**441.88**	3.247	**1.688**
5	468.56	**417.28**	4.085	**1.703**

**Table 2 sensors-21-04604-t002:** Comparison of RMSE and time-consuming of two algorithms on EuRoC dataset.

Sequence	ORB-SLAM	PL-SLAM	DSM	DSO	LDSO	Ours
MH_01_easy	5.13	4.16	**3.9**	4.6	5.3	4.37
MH_02_easy	3.48	5.22	**3.6**	4.6	6.2	3.73
MH_03_medium	7.1	3.99	5.5	17.2	11.4	**3.56**
MH_04_difficult	7.47	6.41	**5.7**	38.1	15.2	6.16
V2_02_medium	6.07	5.65	5.7	13.2	7.8	**5.57**
V2_03_difficult	10.55	12.61	78.4	115.2	×	**9.98**

**Table 3 sensors-21-04604-t003:** Comparison of RMSE (cm) of five algorithms on TUM dataset.

Sequence	ORB-SLAM	LSD-SLAM	RGBD-SLAM	PL-SLAM	Ours
fr1_xyz	1.38	9.00	1.34	1.21	**0.60**
fr1_floor	8.71	38.07	3.51	7.59	**1.76**
fr2_xyz	0.54	2.15	2.61	0.43	**0.29**
fr2_360_kidnap	4.99	×	393.3	3.92	**3.68**
f3_sit_xyz	0.08	7.73	×	0.066	**0.057**

## Data Availability

Not applicable.

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
