# Peer review of "Improved Point-Line Feature Based Visual SLAM Method for Complex Environments"

_sensors, 2021, doi:10.3390/s21134604_

Round 1

Reviewer 1 Report

This manuscript proposed an improved SLAM method from the prospective of point-line feature in the complex environments. The idea is interesting and the result is basically satisfactory. However, some other problems in the manuscript are still concerned in the following:

  1. In the experiments, could the authors compare the proposed method with more state-of-the-art methods to validate the effectivity more extensively?
  2. The organization of this manuscript should be added to the end of the introduction.
  3. The full versions of the abbreviations should be given when they firstly appear in the text.
  4. Most of the references are old. More recent works should be included, such as “DOI: 10.1016/j.eng.2020.03.016”, “DOI: 10.1016/j.isprsjprs.2019.03.002”…
  5. Some references missed the fundamental information, such as volume, number and page. Please check them very carefully.

Author Response

First of all, thank you very much for your careful review and valuable comments. In response to your questions, we have made corresponding explanations and improvements. For details, please see the attachment.

Reviewer 2 Report

This paper uses more environmental structure information by introducing line segments features and design a monocular visual SLAM system. This system combines points and line segments to effectively make up for the shortcomings of traditional positioning based only on point features. Some experiments demonstrate the effectiveness of the proposed method. The main comments are

  1. Please highlight the contribution of this paper for readers to obtain the innovation of this paper.
  2. This paper uses the low-level feature. If the other features obtaining from feature learning method (such as 10.1016/j.neucom.2015.10.035, 3390/rs13071363) can be used for SLAM. Please explain their differences.
  3. In Section 4.2.2, there are many threshold parameters. How to set them should be analyzed in experiments.
  4. Please analyze the reason why the proposed method can achieve better results than the compared methods.
  5. Please discuss the merits and drawbacks of the proposed method.

Author Response

(The authors gave the same response as above.)

Reviewer 3 Report

The paper discusses about the visual simultaneous localization and mapping (SLAM) systems using feature based methods for varying environmental structure by introducing line segments features. 

The paper is well organized and provides adequate results to prove the methods. However, the main problem lies on the presentation as the sections and sub-sections do not match, the fonts are different, the referencing is poor. There are many other feature extraction methods available such as:

Ajay, A., & Venkataraman, D. (2013). A survey on sensing methods and feature extraction algorithms for slam problem. arXiv preprint arXiv:1303.3605.
  Sunny, A. I., Tian, G. Y., Zhang, J., & Pal, M. (2016). Low frequency (LF) RFID sensors and selective transient feature extraction for corrosion characterisation. Sensors and Actuators A: Physical241, 34-43.
  Aulinas, J., Carreras, M., Llado, X., Salvi, J., Garcia, R., Prados, R., & Petillot, Y. R. (2011, June). Feature extraction for underwater visual SLAM. In OCEANS 2011 IEEE-Spain (pp. 1-7). IEEE.   An, S. Y., Kang, J. G., Lee, L. K., & Oh, S. Y. (2010, December). SLAM with salient line feature extraction in indoor environments. In 2010 11th International Conference on Control Automation Robotics & Vision (pp. 410-416). IEEE.   Hartmann, J., Klüssendorff, J. H., & Maehle, E. (2013, September). A comparison of feature descriptors for visual SLAM. In 2013 European Conference on Mobile Robots (pp. 56-61). IEEE.

So the authors should consider different methods of feature extraction used for different applications in order to better derive the solution for the present paper. So it is recommended that these methods are studied and applied. 

Also the tables and figures representing the results need to be explained in the paper. 

The clarity of the figures are important such as Figure 11, 12, 13.

The abstract and conclusion must be supported by the results obtained and not be written in generic form. 

Author Response

(The authors gave the same response as above.)

Reviewer 4 Report

Seen the criticality in the performance when the camera move fact the paper will greatly benefit of more consideration about adding an IMU.

Author Response

(The authors gave the same response as above.)

Round 2

Reviewer 1 Report

All my concerns have been answered.

Reviewer 2 Report

All my concerns have been addressed.